# Severe community-acquired pneumonia in Reunion Island: Epidemiological, clinical, and microbiological characteristics, 2016–2018

Axel de Mangou[1] , Agathe Combe[1], Nathalie Coolen-Allou[2], Guillaume Miltgen[3,4], Nicolas Traversier[3], Olivier Belmonte[3], David Vandroux[1], Michel Bohrer[5], Julien Cousty[6], Margot Caron[1], Charles Vidal[1], Jérôme Allyn[1,7], Nicolas Allou[1,7] *

1 Intensive Care Unit, Centre Hospitalier Universitaire Felix Guyon, Saint-Denis, France, 2 Respiratory Disease, Centre Hospitalier Universitaire Felix Guyon, Saint-Denis, France, 3 Microbiology, Centre Hospitalier Universitaire Felix Guyon, Saint-Denis, France, 4 UMR Processus Infectieux en Milieu Insulaire Tropical, CNRS 9192, INSERM U1187, IRD 249, Université de la Réunion, Saint-Denis, France, 5 Department of Medical Information, Saint-Denis University Hospital, Saint-Denis, Reunion Island, France, 6 Intensive Care Unit, Centre Hospitalier Universitaire Sud Réunion, Saint-Pierre, France, 7 Clinical Informatic Department, Saint-Denis University Hospital, Saint-Denis, Reunion Island, France

* nicolas.allou@hotmail.fr

**Data Availability Statement:** All relevant data are within the paper and its Supporting Information files.

## Abstract

### Purpose

No data are available on severe community-acquired pneumonia (CAP) in the French overseas department of Reunion Island. This is unfortunate as the microorganisms responsible for the disease are likely to differ from those in temperate regions due to a tropical climate and proximity to other islands of the Indian Ocean region. The aim of this study was to assess the epidemiological, clinical, prognosis, and microbiological characteristics of patients with severe CAP in Reunion Island.

### Materials and methods

This retrospective study evaluated all patients with CAP aged >18 years and hospitalized in one of the two intensive care units of Reunion Island between 2016 and 2018. Microorganisms were identified by culture from blood and respiratory samples, multiplex polymerase chain reaction from respiratory samples, urinary antigen tests, and serology.

### Results

Over the study period, 573 cases of severe CAP were recorded, with a mean incidence of 22 per 100,000 person-years. The most frequently isolated microorganism was influenza (21.9%) followed by *Streptococcus pneumoniae* (12%). The influenza virus was detected in affected patients all year round. Twenty-four patients with severe CAP came from another island of the Indian Ocean region (4.2%), mainly Madagascar (>50%). Two of these patients presented with melioidosis and 4 were infected with *Acinetobacter spp*.

**Funding:** The authors received no specific funding for this work.

**Competing interests:** The authors have declared that no competing interests exist.

**Abbreviations:** CAP, Community-acquired pneumonia; ICU, Intensive care unit; PCR, Polymerase chain reaction.

## Conclusions

Our findings have major implications for the management of severe CAP in tropical regions. The most frequently isolated microorganism in patients with severe CAP in Reunion Island is influenza followed by *S. pneumoniae*. Physicians should be aware that influenza is the main cause of severe CAP in patients living in or returning from Reunion Island, where this virus circulates all year round.

## Introduction

Community-acquired pneumonia (CAP) is an acute infection of the lung parenchyma that is acquired outside hospital or health care facilities. It is the most common life-threatening infectious disease. National and international guidelines for the management of severe CAP [1, 2] are based on data collected in regions with a temperate climate, where *Streptococcus pneumoniae*, viruses, and *Legionella* are the main cause of the disease [3, 4]. Accordingly, they recommend to initiate antibiotic therapy effective against all strains of *S. pneumoniae* and *Legionella* (*i.e.*, combination therapy with cephalosporin and a macrolide or monotherapy with respiratory fluoroquinolone) [5] and to consider the possibility of influenza infection during the winter season [6].

These guidelines are also applied in tropical regions, where the microorganisms responsible for the disease are likely to differ from those in temperate regions. This is the case in Reunion Island, a French overseas department with a population of 850,000 inhabitants located in the Indian Ocean region. This tropical island is characterized by two distinct seasons, the hot and humid southern summer from December to May and the milder and drier southern winter the rest of the year. Patients from the entire Indian Ocean region (Madagascar, Mauritius, and the Comoros Archipelago) are regularly evacuated to Reunion, both for reasons of proximity and because the medical infrastructure meets European standards (P3 microbiology laboratories, coronarography, all types of surgeries, circulatory assistance, etc.). This raises the possibility that microorganisms endemic to Madagascar and other neighboring islands, such as *Burkholderia pseudomallei* [7] and *Yersinia pestis* [8], are responsible for some of the cases of severe CAP observed on the island. In spite of this, no comprehensive study of the etiology of severe CAP in Reunion have been conducted to date. A better knowledge of the microbiological characteristics of severe CAP in the region could improve the management of residents or travelers from Reunion Island by helping physicians choose the best antimicrobial treatment according to the season.

The aim of this study was to assess the epidemiological, clinical, prognosis, and microbiological characteristics of patients with severe CAP hospitalized in intensive care unit (ICU) in Reunion Island.

## Materials and methods

### Selection of the study sample

We performed a retrospective chart review of all adult patients diagnosed with severe CAP and hospitalized in one of the two ICUs of Reunion Island (Félix Guyon University Hospital and Saint Pierre University Hospital) between January 2016 and December 2018.

## Ethics

A written notice was provided to all participants or their legally authorized representative. Informed consent was not needed due to the retrospective and non-interventional nature of the study.

This observational study was approved by the French Ethics Committee of Infectious Disease and Tropical Medicine (CER-MIT) and declared to the French National Commission for Data Protection and Liberties (CNIL, #2206739). It complies with the Strengthening the Reporting of Observational studies in Epidemiology recommendations statement [9].

## Definitions

Community-acquired pneumonia was defined as pneumonia acquired outside hospital and diagnosed within 48 hours of hospital admission. Diagnosis was established in the presence of a new lung infiltrate on chest x-ray or computed tomography scan together with one or more of the following symptoms and signs: fever >38˚C, cough, expectoration, chest pain, dyspnea, and signs of invasion of the alveolar space [10].

A severe case of CAP was defined as any patient hospitalized in ICU with 1 of the major criteria and/or 3 or more of the minor criteria established by the American Thoracic Society [1]. Major criteria were septic shock with need for vasopressors and respiratory failure requiring mechanical ventilation. Minor criteria were respiratory rate >30 breaths/min, PaO2/FIO2 ratio <250, multilobar infiltrates, confusion or disorientation, blood urea nitrogen level >20 mg/dL, white blood cell count <4 G/L, platelet count <100 G/L, hypothermia <36˚C, and hypotension requiring aggressive fluid resuscitation.

## Microbiological investigations

Blood and respiratory samples (sputum samples from non-intubated patients and tracheal or bronchoalveolar lavage from intubated patients) were collected from all patients. Microorganism identification was performed on both types of samples using Gram staining followed by culturing for definite identification. Alternatively, identification was performed using matrix-assisted laser desorption ionization time-of-flight mass spectrometry.

Respiratory samples were tested by multiplex polymerase chain reaction (PCR) (Seegene Allplex™ respiratory panel, Eurobio-ingen, Les Ulis, France) for the following microorganisms: influenza, respiratory syncytial virus, adenovirus, enterovirus, parainfluenza, human metapneumovirus, human bocavirus, rhinovirus, coronavirus (NL63, 229E, and OC43), *Chlamydia pneumoniae*, *Mycoplasma pneumoniae*, *Legionella spp*, *Haemophilus influenzae*, *S. pneumoniae*, *Bordetella pertussis*, and *B. parapertussis*.

Pneumococcal and *Legionella* urinary antigen tests were routinely performed on admission to ICU.

Serology for atypical respiratory microorganisms was performed at the physician's discretion.

## Data collection

Patient comorbidities were recorded at hospital admission.

Clinical and biological data were collected at the time of CAP diagnosis.

Average rainfall data for the 2016–2018 period were obtained from *Météo France*, Bureau of Meteorology, Saint-Denis, Reunion Island.

## Study outcome

The primary outcome was to determine the ICU mortality and morbidity of patients with CAP.

The secondary outcome was to identify the microorganisms responsible for CAP in patients hospitalized in ICU.

## Statistical analysis

Results were expressed as total number (percentage) for categorical variables and as median [25th-75th percentiles] for continuous variables. Continuous variables were compared using the Mann-Whitney test and categorical variables using the Chi-square test or Fisher's exact test, as appropriate. Survival functions in ICU at 30 days were estimated using the Kaplan-Meier method and compared using the log-rank test. A *P* value <0.05 was considered significant. Analyses were performed using the SAS statistical software (8.2, Cary, NC, USA).

# Results

## Incidence and isolated microorganisms

From January 2016 to December 2018, 1,283 patients were admitted to ICU for suspected or confirmed lower tract respiratory infection. Of these, 710 patients were excluded (12 were <18 years old, 698 had nosocomial pneumonia or non-infectious pneumonia). The remaining 572 patients formed the study cohort. The mean incidence of severe CAP was 22 per 100,000 person-years. The median age was 62 [52–73] years and the median simplified acute physiology *score* II on admission was 44 [31–57]. Patients presented with acute respiratory distress syndrome in 396 cases (69.2%) and with sepsis or septic shock in 347 cases (60.7%). Patient characteristics at ICU admission are shown in Tables 1 and 2.

The microorganism(s) responsible for severe CAP were identified in 67% of cases. The most frequently isolated microorganisms were influenza (21.9%), *S. pneumoniae* (12%), *Staphylococcus* spp (10.8%), *Enterobacteriaceae* (9.8%), and *H. influenzae* (7.5%). Panton-Valentine leukocidin-positive *Staphylococcus aureus* accounted for 11.3% of all *Staphylococcus spp* strains. *Legionella pneumophila* and other intracellular bacteria were responsible for 3% of cases (Table 3). Other isolated viruses were rhinovirus in 10 cases (1.7%), human metapneumovirus in 9 cases (1.6%), coronavirus OC43 in 9 cases (1.6%), parainfluenza in 9 cases (1.6%), respiratory syncytial virus in 8 cases (1.4%), and adenovirus in 2 cases (0.3%). In the group of patients with influenza CAP, infection was polymicrobial in 52 cases (41.6%).

The incidence of influenza vs. *S. pneumoniae* according to rainfall is shown in Fig 1.

Twenty-four patients with severe CAP came from another island of the Indian Ocean region (4.2%), mainly Madagascar (>50%). Two of these patients presented with melioidosis (which is caused by *B. pseudomallei*) and 4 were infected with *Acinetobacter* spp (2 with *Acinetobacter baumannii* and 2 with *A. pittii)*.

## Prognosis

Over the study period, ICU mortality for the entire cohort was 20.8%. Survival rates according to microorganism are shown in Fig 2.

In univariate analysis, mortality was higher in patients with influenza CAP (24%) than in patients with non-influenza CAP (18.9%, *P* = 0.04).

The median duration of ICU stay was 7 [4–17] days in patients with influenza CAP vs. 6 [3–11] days in patients with non-influenza CAP (*P* = 0.018).

**Table 1. Clinical characteristics at intensive care unit admission.**

| Clinical characteristics | Missing data | Total | Influenza CAP | Non-Influenza CAP | P-value |
|---|---|---|---|---|---|
| | | (n = 572) | (n = 125) | (n = 447) | |
| Age (years) | 0 | 62 [52–73] | 61 [48.3–69] | 63 [52.2–74] | 0.099 |
| Male | 0 | 376 (65.7) | 73 (58.4) | 303 (67.8) | 0.013 |
| Body mass index (kg/m$^2$) | 45 | 24.1 [21.4–29] | 25.47 [22–29.9] | 23.8 [21.1–28.63] | 0.079 |
| [1]Foreign residence | 0 | 26 (4.5) | 3 (2.4) | 23 (5.1) | 0.091 |
| Duration of symptoms before ICU admission (days) | 0 | 2 [1–5] | 3 [2–6] | 2 [1–5] | 0.018 |
| [2]CURB-65 score | 8 | 3 [2–3] | 3 [2–4] | 3 [2–3] | 0.122 |
| SAPS II | 12 | 53 [28–68] | 55 [37–58.5] | 48.5 [28.7–68.7] | 0.795 |
| Pulmonary abscess | 0 | 38 (6.6) | 7 (5.6) | 31 (6.9) | 0.147 |
| Influenza-like illness | 0 | 175 (30.6) | 68 (54.4) | 107 (23.9) | <0.001 |
| Immunosuppression | 0 | 47 (8.2) | 9 (7.2) | 38 (8.5) | 0.136 |
| Corticosteroids | 0 | 53 (9.3) | 7 (5.6) | 46 (10.3) | 0.04 |
| [3]Hazardous alcohol use | 3 | 159 (27.8) | 26 (20.8) | 133 (29.8) | 0.13 |
| Chronic obstructive pulmonary disease | 0 | 144 (25.2) | 26 (20.8) | 118 (26.4) | 0.042 |
| Asthma | 0 | 33 (5.8) | 13 (10.4) | 20 (4.5) | 0.009 |
| Hypertension | 0 | 248 (43.3) | 56 (44.8) | 192 (43) | 0.076 |
| Chronic renal failure with dialysis | 0 | 28 (4.9) | 5 (4) | 23 (5.1) | 0.171 |
| Diabetes mellitus | 0 | 197 (34.4) | 48 (38.4) | 149 (33.3) | 0.048 |
| Liver cirrhosis | 0 | 22 (3.8) | 4 (3.2) | 18 (4.0) | 0.2 |
| Cancer < 4 months | 0 | 51 (8.9) | 6 (4.8) | 45 (10.1) | 0.026 |
| History of congestive heart failure | 0 | 114 (19.9) | 18 (14.4) | 96 (21.5) | 0.022 |
| Pregnancy | 0 | 7 (1.2) | 2 (1.6) | 5 (1.1) | 0.294 |

Results are expressed as n (%) or median [25th-75th percentiles], as appropriate.

CAP: Community-acquired pneumonia; ICU: Intensive care unit; SAPS 2: Simplified acute physiology score 2 [11].

[1]Wounded and sick patients transported from foreign countries.

[2]CURB-65 (confusion: 1 point; blood urea nitrogen>19mg per dl:1 point; respiratory rate>30:1 point; systolic blood pressure<90mmHg and/or diastolic blood pressure<60mmHg:1 point; age>65 years:1 point) [12].

[3]Alcohol Use Disorders Identification Test Consumption ≥ 4 for men or 3 for women [13].

The median duration of mechanical ventilation was 4 [0–14] days in patients with influenza CAP vs. 2 [0–8] days in patients with non-influenza CAP (*P* = 0.04).

The need for extracorporeal membrane oxygenation was 8% in patients with influenza CAP vs. 2.5% in patients with non-influenza CAP (*P* = 0.005).

## Discussion

This is the first epidemiological study to assess the clinical, microbiological, and prognostic characteristics of severe CAP in Reunion Island. It is also one of the rare studies on CAP in the Indian Ocean region [14, 15]. The incidence of severe CAP in our cohort was 22 per 100,000 person-years. The most frequently isolated microorganism was influenza (21.9%) followed by *S. pneumoniae* (12%). These epidemiological data can help to implement appropriate anti-infective treatment in residents or travelers from Reunion Island with severe CAP [16].

The rate of microbiological identification in our study was relatively high (68.3%) compared to other studies on the subject (around 50%) [3, 17–20]. This difference may be explained by our use of multiplex PCR, which allowed to detect both viral and bacterial agents of the disease. As in our study, recent studies using PCR for microbiological identification [20–22] found viruses (and especially influenza) to be the most common cause of severe CAP.

**Table 2. Prognostic factors and laboratory findings at intensive care unit admission.**

| Variables | Missing data | Total (n = 572) | Influenza CAP (n = 125) | Non-Influenza CAP (n = 447) | *P*-value |
|---|---|---|---|---|---|
| **Prognostic factors** | | | | | |
| Temperature (°C) | 23 | 38 [36.6–38.7] | 38.3 [37.1–39] | 37.7 [36.5–38.6] | 0.017 |
| Glasgow Coma Scale score | 1 | 15 [3–15] | 15 [3–15] | 15 [3–15] | 0.348 |
| Extracorporeal membrane oxygenation | 0 | 21 (3.7) | 10 (8) | 11 (2.5) | 0.005 |
| Mechanical ventilation | 0 | 325 (56.8) | 76 (60.8) | 249 (55.7) | 0.049 |
| Non-invasive ventilation | 0 | 122 (21.3) | 31 (24.8) | 91 (20.4) | 0.054 |
| High-flow oxygen therapy | 0 | 72 (12.6) | 17 (13.6) | 55 (12.3) | 0.11 |
| Renal replacement therapy | 0 | 74 (12.9) | 19 (15.2) | 55 (12.3) | 0.08 |
| Catecholamines | 0 | 283 (49.5) | 64 (51.2) | 219 (49) | 0.073 |
| Pa02/FiO2 ratio (mmHg) | 6 | 166 [110.25–240] | 152 [97.5–205.5] | 179.5 [115–252.8] | 0.003 |
| **Laboratory findings** | | | | | |
| Creatinine level (μmol/L) | 0 | 109 [70–178] | 106 [70–148.25] | 110 [71.25–187] | 0.335 |
| Total bilirubin level (μmol/L) | 12 | 10 [6–16] | 9 [6–15] | 11 [6–16] | 0.414 |
| Prothrombin time (%) | 8 | 70 [55–83] | 72.5 [58.75–87] | 67 [53–81] | 0.029 |
| Platelet count (G/L) | 1 | 188 [129–271] | 158 [119–218] | 203.5 [136–285.5] | <0.001 |
| Leucocyte count (G/L) | 0 | 11.7 [7.5–16.3] | 8.9 [5.5–14.5] | 12.1 [8.15–16.65] | <0.001 |
| Lactate level (mmol/L) | 9 | 1.95 [1.3–3.4] | 2 [1.3–3.2] | 1.9 [1.3–3.5] | 0.859 |
| Creatine phosphokinase (mg/dL) | 18 | 210 [87–604] | 260 [122–947] | 184 [83–513] | 0.036 |
| Hemoglobin level (g/dL) | 0 | 12 [10.1–13.6] | 12.75 [11.25–14] | 11.76 [10–13,5] | 0.001 |
| Alanine aminotransferase level (UI/L) | 11 | 28 [17–51.25] | 32 [16–55] | 26 [17–51] | 0.271 |
| Troponin level (ng/dL) | 24 | 37 [16–104] | 25 [11–82] | 38 [18–116.25] | 0.017 |
| C-reactive protein level (mg/L) | 54 | 170 [73.25–304.25] | 148 [71.5–250.5] | 177.5 [73.1–324] | 0.558 |

Results are expressed as n (%) or median [25th-75th percentiles], as appropriate.

**Table 3. Isolated microorganisms.**

| *Microorganisms* | Total (n = 572) | Influenza CAP (n = 125) | Non-Influenza CAP (n = 447) |
|---|---|---|---|
| Viruses | 168 (29.3) | 125 (100) | 43 (9.6) |
| Bacteria | 283 (49.5) | 50 (40) | 233(52.1) |
| *Staphylococcus spp* | 62 (10.8) | 24 (19.2) | 38 (8.5) |
| Panton-Valentine Leukocidin-positive | 7 (1.2) | 4 (3.2) | 3 (0.7) |
| *Streptococcus pneumoniae* | 69 (12) | 16 (12.8) | 53 (11.8) |
| *Haemophilus influenzae* | 43 (7.5) | 5 (4) | 38 (8.5) |
| Enterobacteriaceae | 56 (9.8) | 3 (2.4) | 53 (11.8) |
| Non-fermenting gram-negative bacilli | 26 (4.5) | 2 (1.6) | 24 (5.4) |
| *Legionella* | 9 (1.5) | 0 | 9 (2) |
| Intracellular | 9 (1.5) | 0 | 9 (2) |
| Fungi | 8 (1.4) | 2 (1.6) | 6 (1.3) |

Results are expressed as *n* (%).

Polymicrobial infection was found in 52 cases of influenza CAP (41.6%).

Bacteremia was found in 97 cases of severe CAP (16.9%).

Bacteremia was found in 44 cases of influenza CAP (35.2%) and in 53 cases of non-influenza CAP (11.8%).

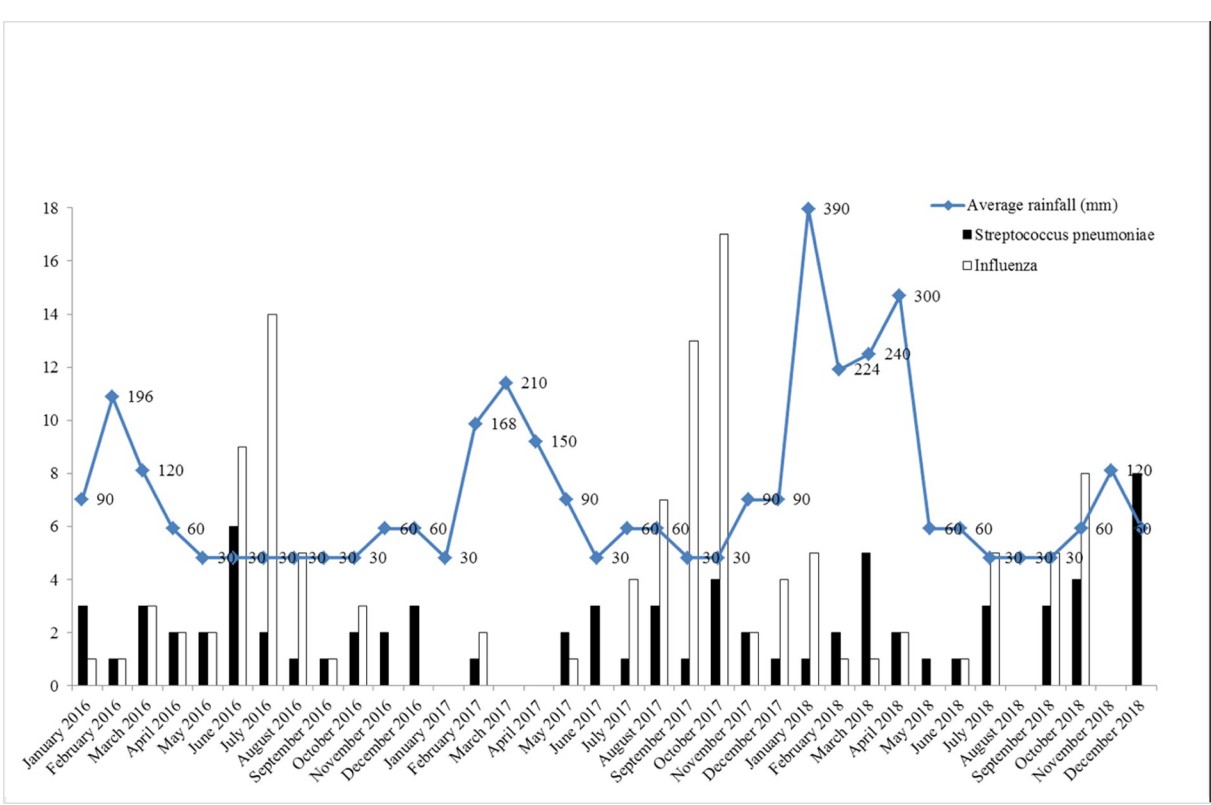

**Fig 1. Incidence of influenza vs. *S. pneumoniae* according to rainfall, 2016 to 2018, Reunion Island.**

Thus, in the 2016 retrospective study conducted by Visseaux *et al.* in mainland France [21], 29.2% of 5,000 ICU patients with severe CAP had a viral infection. This finding is in line with our study, which detected viruses in 29.3% of patients with severe CAP. By contrast, the influenza virus accounted for 74.7% of all detected viruses in our study compared to 34.4% in the study by Visseaux *et al.* Siow *et al.* [23] have stressed the importance of using PCR for the detection of viruses causing severe CAP in tropical environments, where the incidence of viral infections is less dependent on the seasons than in temperate regions [24, 25]. In our study, this approach showed that influenza circulates all year round in Reunion Island. In spite of this, peaks of influenza incidence were observed during the southern winter and monthly incidence was found to be inversely proportional to temperature and rainfall. Several other studies conducted in tropical regions have found an association between seasons and the occurrence of influenza infections [25–29]. However, the physio-pathological explanation for this association remains to be established [24].

In our cohort, *S. pneumoniae* was the second (12%) most frequently isolated microorganism. This contrasts with most available studies, which identify *S. pneumoniae* as the main cause of severe CAP. However, many of these studies do not provide the incidence of viral infections [1, 2, 5, 18, 23] or underestimate it due to the non-systematic use of PCR for microorganism detection. In the last epidemiological study on severe CAP in Reunion Island, Paganin *et al.* [18] identified *S. pneumoniae* (42.9%) and *Klebsiella pneumoniae* (22.4%) as the main cause of the disease. Yet, this 2004 study did not establish the viral etiology of severe CAP because multiplex PCR was not systematically used in local ICUs at the time.

In our study, 2 patients returning from Madagascar presented with melioidosis. Given the recent emergence of *B. pseudomallei* in the Indian Ocean islands [7], clinicians and

## Survival functions

**Fig 2. Survival rate according to microorganism using Kaplan–Meier analysis.**

microbiologists should consider melioidosis as a differential diagnosis in patients returning from the region, in particular from Madagascar. Other atypical microorganisms were detected, namely 2 strains of *A. baumannii* and 2 strains of *A. pittii*, with a mortality of 50%. While *Acinetobacter spp* strains are mainly known as nosocomial infectious agents, they have been shown to cause severe CAP in tropical areas. Thus, an observational study [30] conducted in Australia reported 41 cases of CAP caused by *Acinetobacter spp*, with *A. baumannii* accounting for 85% of strains. In that study, 88% of infectious episodes occurred during the rainy season, and 80% of patients required hospitalization in ICU.

The mortality rate for our entire cohort was 20.8%. This figure is consistent with those reported in the recent studies by Ferrer *et al.* (22%) [31] and Cavallazzi *et al.* (27%) [32]. By contrast, a UK study found mortality to reach 50% in patients with CAP hospitalized in ICU [33]. In the study conducted by Dupuis *et al.* in mainland France, hospital mortality was 22.8% in a cohort of 1,665 patients with severe pneumococcal CAP [34]. In our study, mortality and morbidity (including the need for extracorporeal membrane oxygenation and mechanical

ventilation) were higher in patients with influenza CAP than in those with non-influenza CAP. Likewise, in the Spanish study by Abelleira *et al*., patients with CAP caused by influenza A H1N1 had a poorer prognosis than those with non-influenza CAP [35]. In a study conducted during the 2009 AH1N1 pandemic, Vandroux *et al*. [36] found mortality and the need for extracorporeal membrane oxygenation to be extremely high in Reunionese patients with AH1N1 influenza. Interestingly, Vandroux *et al*. reported numerous co-infections (31%), nearly half of which were caused by influenza and *S. aureus*. This is consistent with our results, since 41.6% of influenza infections were polymicrobial in our cohort, with half of them involving *S. aureus*. Other studies have found that mixed viral-bacterial infections are associated with an increased risk of mortality in patients with severe CAP [37, 38].

In line with the recommendations of the Infectious Diseases Society of America and the World Health Organization, the French Society of Infectious Disease and the French National Authority for Health recommend initiating curative treatment with oseltamivir in patients at risk of severe influenza [39]. Several recent studies confirm the potential benefit of early treatment with oseltamivir [40–43].

Our study has many limitations. First, the retrospective nature of the study could have introduced biases in our results. Second, only patients with severe CAP were included in the sample, which means that our results do not reflect the exact etiological agents of CAP in Reunion Island. Third, since our study covers the 2016–2018 period, it does not account for changes in the ecology of severe CAP that likely resulted from the COVID-19 pandemic and the implementation of social distancing measures [44–46]. Lastly, in the absence of multivariate analysis, we cannot discard the possibility that the difference in mortality rates observed between patients with influenza CAP vs. non-influenza CAP is due to the fact that the first group of patients had more comorbidities (age over 65 years, asthma, diabetes mellitus, etc.).

## Conclusion

Our findings have major implications for the management of severe CAP in tropical regions. The most frequently isolated microorganism in patients with severe CAP in Reunion Island is influenza followed by *S. pneumoniae*. Physicians should be aware that influenza is the main cause of severe CAP in patients living in or returning from Reunion Island, where this virus circulates all year round.

## Supporting information

**S1 Dataset.**
(XLS)

## Author Contributions

**Conceptualization:** Axel de Mangou, Agathe Combe, Nathalie Coolen-Allou, Guillaume Miltgen, Nicolas Traversier, Olivier Belmonte, David Vandroux, Margot Caron, Charles Vidal, Jérôme Allyn, Nicolas Allou.

**Data curation:** Axel de Mangou, Agathe Combe, Guillaume Miltgen, Olivier Belmonte, David Vandroux, Michel Bohrer, Charles Vidal, Nicolas Allou.

**Formal analysis:** Axel de Mangou, Agathe Combe, Guillaume Miltgen, David Vandroux, Julien Cousty, Margot Caron, Nicolas Allou.

**Funding acquisition:** Axel de Mangou, Jérôme Allyn.

**Investigation:** Axel de Mangou, Agathe Combe, Nathalie Coolen-Allou, Olivier Belmonte, Jérôme Allyn, Nicolas Allou.

**Methodology:** Agathe Combe, Nathalie Coolen-Allou, Charles Vidal, Jérôme Allyn, Nicolas Allou.

**Project administration:** Axel de Mangou.

**Resources:** Michel Bohrer.

**Supervision:** Nicolas Traversier, Charles Vidal, Jérôme Allyn, Nicolas Allou.

**Validation:** Nicolas Traversier, Margot Caron, Charles Vidal, Jérôme Allyn, Nicolas Allou.

**Visualization:** Nicolas Traversier, Julien Cousty, Margot Caron, Nicolas Allou.

**Writing – original draft:** Charles Vidal, Jérôme Allyn, Nicolas Allou.

**Writing – review & editing:** Jérôme Allyn, Nicolas Allou.

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
