## [Decision Letter · Decision Letter 0]

23 Jul 2021

PONE-D-21-05049

Severe community-acquired pneumonia in Reunion Island:Epidemiological, clinical and microbiological characteristics, 2016-2018

PLOS ONE

Dear Dr. Allou,

Thank you for submitting your manuscript to PLOS ONE. After careful consideration, we feel that it has merit but does not fully meet PLOS ONE’s publication criteria as it currently stands. Therefore, we invite you to submit a revised version of the manuscript that addresses the points raised during the review process.

While your paper addresses an interesting question, both reviewers expressed significant concern both about the presentation as well as the readability of the manuscript and did not recommend publication in its present form. In particular, both reviewers voice a number of concerns regarding the used methodology for data collection and analysis, and these comments need to be addressed carefully.  Please see reviewers’ insightful comments below. In addition, the quality of the language needs to be improved, please have a fluent, preferably native, English-language speaker thoroughly copyedit your manuscript for language usage, spelling, and grammar.

Personally, I found the following points need to be clarified:  1. Prevalent pathogens for CAP are influenza, *S. penumoniae*. How does this compare to nosocomial pneumonia? 2. CAP caused by influenza and *S. penumoniae* is year round, however, the authors indicated the rainfall play a role in the case of CAP.  From the figure, it appears more case for influenza CAP in dry season. Any studies on influenza CAP related to the same phenomenon? 3. The mortality rate is higher for influenza CAP, is this same from other region?

Specific comments:

Line 54, 97 “All consecutive patients…” I am not sure consecutive is the right adj. to use.  Suggest changing to “All patients…”Line 79 – 81, is there any CAP data in nearby area, Madagascar for example?Line 79, 81 &118, change “legionella” to “*Legionella*” since genus used alone should be capitalized and italicized.Line 82,153, 194, 233 & 301,*Streptococcus pneumonia *should be* “S. pneumoniae”*Line 116 – 117, change “*Chlamydia*
*pneumonia*, *Mycoplasma pneumonia*, *Legionella spp*, *Haemophilus influenza*, *Streptococcus pneumonia*e and *Bordetella (para) pertussis.*” to “*Chlamydia pneumoniae*, *Mycoplasma pneumoniae*, *Legionella spp*, *Haemophilus influenzae*, *S. pneumoniae*, *Bordetella pertussis, *and* B. prarpertussis.”*Line 142, please rephrase “…(incidence of 22 per 100,000 person- years)” for clarity.  Did the author mean 22 per 100,000 person per year?Line 154, Staphylococcus should be italicized.Line 155, “*Haemophilus influenza*” should be “*H. influenzae*”Line 155 – 156, “*Legionella pneumophila*” should be “*L. pneumophila*”Table 3, “*Haemophilus influenza*” should be “*Haemophilus influenzae*”Line 193 – 195, it is not clear about the correlation between rainfall and CAP, please expand the explanation.Line 200 – 206, maybe I missed something, I am not clear on the relevance of the risk factors for influenza with CAP?  How about risk factor associated with *S. pneumoniae *or CAP in general?Line 220 – 222, please clarify the relevance of this statement.Line 248, change “*K. pneumoniae*” to “*Klebsiella pneumonia*” since this is the first time *Klebsiella* was mentioned.Line 274, Actinobacter should be italicized.Line 285, “*Staphylococcus aureus*” should be “*S. aureus”*

We look forward to receiving your revised manuscript.

Kind regards,

Baochuan Lin, Ph.D.

Academic Editor

PLOS ONE

Journal Requirements:

2. We note that you obtained consents from participants to take part in your retrospective study. In the Ethics Statement on the online submission form and the manuscript Methods , please clarify the context in which consent was obtained, and specify whether patients provided:

                a) Consent to use their medical records/samples in research

                b) Consent to undergo the procedure

                c) Consent to take part in the study reported in this manuscript.

If the ethics committee waived the need for additional informed consent, please state this.

Thank you for your attention to these requests.

Reviewers' comments:

Reviewer's Responses to Questions

**Comments to the Author**

1. Is the manuscript technically sound, and do the data support the conclusions?

Reviewer #1: Partly

Reviewer #2: Partly

2. Has the statistical analysis been performed appropriately and rigorously? 

Reviewer #1: Yes

Reviewer #2: No

3. Have the authors made all data underlying the findings in their manuscript fully available?

Reviewer #1: No

Reviewer #2: Yes

4. Is the manuscript presented in an intelligible fashion and written in standard English?

Reviewer #1: Yes

Reviewer #2: Yes

5. Review Comments to the Author

Reviewer #1: Dear Editor PLOS ONE,

Thank you for sending me this manuscript for review. The investigators studied clinical and microbiological characteristics of patients with severe pneumonia who were admitted to the ICUs. The manuscript is of interest, however, several points need to be highlighted and revised as follow.

Abstract

Editing error: “causatives”

Conclusions are not based on the results. Also, the recommendation of cephalosporin and oseltamivir needs revision to reflect the study findings and may lead to increased antimicrobial resistance.

Introduction

Some general statements require evidence and to be specific, for example:

Initiate antibiotic therapy against all strains of S pneumonia and Legionella. This is a general statement that needs to be more specific on the empiric use of antimicrobials.

Methods

The study was conducted on ICU admitted patients, and they may not represent the study objectives rather than only severe cases that required ICU admission. So, study objectives need to be changed accordingly.

It’s not clear too, how data were collected, was it a retrospective chart review?

Statistical analysis

Line 5: First time mentioned abbreviation, the full name should be provided.

Also, the statement is not clear: “Risk factors of influenza PAC in bivariate analysis with P<0.1 were entered into a multivariate logistic regression analysis using backward selection with P <0.05”.

The reported incidence in the results section was different from the abstract and discussion, should be revised.

Table (1), abscess was mentioned, what does it refer to what type of abscess and which site?

Table (1), duration between symptoms and ICU admission, was it days?

Table (1), what does it mean of reporting the p-value as zero? (for influenza-like illness)

Table (1), how hazardous alcohol use was assessed, what definition was used, and how data were collected?

Table (1), what does footnote 2 refer to?

Did the investigators collect any vaccination histories on influenza or pneumococcal vaccines among the study participants?

Table (3), footnote … “…. during flu cases” need revision.

What is “flu syndrome” mentioned in line 201?

Higher leukocyte counts mentioned in line 204 is not a risk for severe influenza as per the odds ratio presented.

Line 212 need revision

Discussion

The second highest cause of mortality and years of life lost. Was there a difference between mortality and years of life lost?

in-ICU mortality in this study was lower than previously reported. Any specific reasons or contributing factors to be discussed?

Editing, line 253, “….. we find” should “found”

What about the impact of empirical use of antimicrobials on the detection of organisms that have been reported in other reports.

The investigators included a paragraph in the discussion about climate change and influenza types, although they didn’t provide a specific discussion or explored the relevance to the study.

Conclusions should be revised to reflect the study findings.

Also, the recommendation of probabilistic antimicrobials and oseltamivir to every patient should be revised taking into consideration the potential risk for antimicrobial resistance and definition of severe pneumonia that may not be specific to be applicable in several care settings.

Figure (2), worst survival was observed for cases with S pneumonia, this needs further discussion

Figure (1) (which is also written as figure (2)), about rainfall and influenza, showed three peaks of rainfall but not clear of an association with the occurrence of influenza or pneumococcal infection, more elaboration in this regard is required.

Reviewer #2: Severe community-acquired pneumonia in Reunion Island: Epidemiological, clinical and microbiological characteristics, 2016-2018 (Manuscript No. PONE-D-21-05049)

In their manuscript, de Mangou et al. present valuable data derived from a retrospective cohort of patients admitted and diagnosed with severe CAP at Reunion Island between 2016 and 2018. However, significant scarcities of methodology detailed below could be identified in the manuscript. Also, the main study results might be of more interest to the national readers of Reunion Island. In all, the Methods section needs rewriting by addition of more explicit details from the study protocol before the review process could be continued any further.

Remarks on methodology:

1. A major issue is that CAP was not defined properly (only criteria of severe CAP were stated), and details were not given on how cases of CAP were ascertained. A brief summary of in-hospital algorithms should be added to the manuscript. Also, did all the patients receive the same diagnostic (eg. imaging) and microbiological studies before the final diagnosis of CAP was arrived at?

2. What is a cytobacteriological examination? This needs further explaining.

3. Did the patients have blood cultures taken?

4. Are all patients with severe CAP hospitalized at the ICU ward at Reunion Island?

5. How were eligible patients screened and included in the final cohort?

6. Was any patient follow-up seeked during the study period? If yes, how?

7. How was data collection executed? What was the data source? How were missing data handled?

8. The primary outcome should be defined by a hard clinical end-point, eg. in-hospital all-cause mortality at day-14 or alike. "Clinical and microbiological characteristics" are not endpoints – they are characteristics.

9. There is a contradiction in the description of statistical methods: an univariate pre-screening was done (line 133), but after that, only the "most clinically relevant factors" were entered into multivariate analysis (line 135)? This needs to be elaborated.

10. Adherence to the STROBE Statement is highly encouraged during rewriting.

11. I suggest using the term bronchoalveolar lavage instead of bronchoalveolar sputum.

12. I suggest that the term "adult patients" should be emphasized.

13. The name of the American Thoracic Society should be corrected.

6. PLOS authors have the option to publish the peer review history of their article (what does this mean?). If published, this will include your full peer review and any attached files.

Reviewer #1: No

Reviewer #2: No

---

## [Author Response · Author response to Decision Letter 0]

4 Nov 2021

Nicolas Allou, MD August, 27th, 2021

CHU Félix Guyon

Réanimation Polyvalente

Allée des Topazes, 97400, Saint Denis, France

Phone: +2 62 90 56 94

Fax: +2 62 90 66 93

E-mail: nicolas.allou@hotmail.fr
nicolas.allou@chu-reunion.fr

Dear Pr. Baochuan Lin,

Thank you for your letter concerning our manuscript entitled “Severe community-acquired pneumonia at Reunion Island: Epidemiological, clinical and microbiological characteristics, 2016-2018" (PONE-D-21-05049).

The comments were stimulating and challenging. We have therefore deeply modified the manuscript, taking all of the suggestions into account. Changes in the manuscript are in red font.

The manuscript has not and will not be offered elsewhere for possible publication, as long as it is under PLOS ONE consideration.

Ethics statement and consents from participants have been specified in the revised manuscript

We have added the data set of the study

We give a point-by-point response to the reviewers (see below) 

We hope that these revisions have improved the manuscript.

Nicolas ALLOU, MD 

COMMENTS TO AUTHOR:

Personally, I found the following points need to be clarified: 

1. Prevalent pathogens for CAP are influenza, S. penumoniae. How does this compare to nosocomial pneumonia?

We have better redefined our diagnosis criteria for CAP in the material and methods section (lines 108-112).

2. CAP caused by influenza and S. penumoniae is year round, however, the authors indicated the rainfall play a role in the case of CAP. From the figure, it appears more case for influenza CAP in dry season. Any studies on influenza CAP related to the same phenomenon?

Several studies find an association between seasons and the occurrence of CAP and more particularly concerning influenza (Radina P, et al. Associations between seasonal influenza and meteorological parameters in Costa Rica, Honduras and Nicaragua. Geospat Health 2015 Nov 4;10(2):372/Oyelola A et al. Epidemiological analysis of association between lagged meteorological variables and pneumonia in wet-dry tropical North Australia, 2006-2016. J Expo Sci Environ Epidemiol 2020 May;30(3):448-458/ Julian W Tang, et al. Comparison of the incidence of influenza in relation to climate factors during 2000-2007 in five countries. J Med Virol 2010 Nov;82(11):1958-65/ Gwladys C, et al. Associations between meteorological parameters and influenza activity in a subtropical country: Case of five sentinel sites in Yaoundé-Cameroon. PLoS One 2017 Oct 31;12(10):e0186914/ Kramer SC. Development and validation of influenza forecasting for 64 temperate and tropical countries. . PLoS Comput Biol 2019 Feb 27;15(2):e1006742/Gentile A, et al. Influenza virus: 16 years' experience of clinical epidemiologic patterns and associated infection factors in hospitalized children in Argentina. PLoS One 2018 Mar 29;13(3):e0195135/Cilloniz C et al. Seasonality of pathogens causing community-acquired pneumonia. Respirology 2017 May;22(4):778-785). It has been added in the revised manuscript (lines 247-249)

3. The mortality rate is higher for influenza CAP, is this same from other region?

Mortality rate was higher for influenza CAP than for non-influenza CAP. The impact on mortality of the pathogen causing CAP is variable according to the studies. In Spain, Abelleira et al. found that CAP related to influenza A H1N1 had a poorer prognosis than non-influenza CAP (Abelleira R, Ruano-Ravina A, Lama A, Barbeito G, Toubes ME, Domínguez-Antelo C, González-Barcala FG, Rodríguez-Núñez N, Marcos PJ, Del Molino ML, Valdés L. Influenza A H1N1 Community-Acquired Pneumonia: Characteristics and Risk Factors-A Case-Control Study. Can Respir J 2019 Mar 17;2019:4301039. doi: 10.1155/2019/4301039. eCollection 2019) (Quah J, Jiang B, Tan PC, Siau C, Tan TY. Impact of microbial Aetiology on mortality in severe community-acquired pneumonia. BMC Infect Dis. 2018 Sep 4;18(1):451. doi: 10.1186/s12879-018-3366-4). It has been also found that mixed viral-bacterial infections may be associated with an increased risk of mortality (Quah J, Jiang B, Tan PC, Siau C, Tan TY. Impact of microbial Aetiology on mortality in severe community-acquired pneumonia. BMC Infect Dis. 2018 Sep 4;18(1):451. doi: 10.1186/s12879-018-3366-4)

It has been added in the revised manuscript (line 274-278).

Specific comments:

1. Line 54, 97 “All consecutive patients…” I am not sure consecutive is the right adj. to use. Suggest changing to “All patients…”

We agree with reviewer comment, it has been corrected in the revised manuscript.

2. Line 79 – 81, is there any CAP data in nearby area, Madagascar for example?

There are only few studies on CAP in the Indian Ocean region. There are only 2 studies concerning Madagascar but none concerning Mauritius or the Comoros archipelago (Razanajatovo NH, Guillebaud J, Harimanana A, Rajatonirina S, Ratsima EH, Andrianirina ZZ, Rakotoariniaina H, Andriatahina T, Orelle A, Ratovoson R, Irinantenaina J, Rakotonanahary DA, Ramparany L, Randrianirina F, Richard V, Heraud JM. Epidemiology of severe acute respiratory infections from hospital-based surveillance in Madagascar, November 2010 to July 2013. PLoS One. 2018 Nov 21;13(11):e0205124. doi: 10.1371/journal.pone.0205124. eCollection 2018) (J L Rakotoson, J R Rakotomizao, A C F Andrianarisoa. Acute community acquired pneumonia: 96 cases in Madagascar . Med Trop (Mars) 2010 Feb;70(1):62-4).

It has been added in the revised manuscript (line 217).

3. Line 79, 81 &118, change “legionella” to “Legionella” since genus used alone should be capitalized and italicized. 

We apologize, it has been corrected in the revised manuscript 

4. Line 82,153, 194, 233 & 301,Streptococcus pneumonia should be “S. pneumoniae”

We apologize, it has been corrected in the revised manuscript

5. Line 116 – 117, change “Chlamydia pneumonia, Mycoplasma pneumonia, Legionella spp, Haemophilus influenza, Streptococcus pneumoniae and Bordetella (para) pertussis.” to “Chlamydia pneumoniae, Mycoplasma pneumoniae, Legionella spp, Haemophilus influenzae, S. pneumoniae, Bordetella pertussis, and B. prarpertussis.”

We apologize, it has been corrected in the revised manuscript

6. Line 142, please rephrase “…(incidence of 22 per 100,000 person- years)” for clarity. Did the author mean 22 per 100,000 person per year?

We agree with reviewer comment it has been modified in the revised manuscript

7. Line 154, Staphylococcus should be italicized.

8. Line 155, “Haemophilus influenza” should be “H. influenzae”

9. Line 155 – 156, “Legionella pneumophila” should be “L. pneumophila”

10. Table 3, “Haemophilus influenza” should be “Haemophilus influenzae”

We apologize, it has been corrected in the revised manuscript

11. Line 193 – 195, it is not clear about the correlation between rainfall and CAP, please expand the explanation.

We agree with reviewer comment. Several studies find an association between seasons and the occurrence of CAP but the physio-pathological explanation of this association remains to be established. This explanatory paragraph was removed because it was unclear

12. Line 200 – 206, maybe I missed something, I am not clear on the relevance of the risk factors for influenza with CAP? How about risk factor associated with S. pneumoniae or CAP in general?

We agree with reviewer comment, this analysis has been removed from the revised manuscript. Risk factors for developing CAP could not be assessed because there was no control group (without CAP).

13. Line 220 – 222, please clarify the relevance of this statement.

We agree with reviewer comment this section has been removed from the revised manuscript

14. Line 248, change “K. pneumoniae” to “Klebsiella pneumonia” since this is the first time Klebsiella was mentioned.

15. Line 274, Actinobacter should be italicized.

16. Line 285, “Staphylococcus aureus” should be “S. aureus”

We apologize, it has been corrected in the revised manuscript

Reviewer #1: 

Thank you for sending me this manuscript for review. The investigators studied clinical and microbiological characteristics of patients with severe pneumonia who were admitted to the ICUs. The manuscript is of interest, however, several points need to be highlighted and revised as follow.

Abstract

Editing error: “causatives”

We agree with reviewer comment, it has been corrected in the revised manuscript.

Conclusions are not based on the results. Also, the recommendation of cephalosporin and oseltamivir needs revision to reflect the study findings and may lead to increased antimicrobial resistance.

We agree with reviewer comment, it has been corrected in the revised manuscript.

Introduction

Some general statements require evidence and to be specific, for example:

Initiate antibiotic therapy against all strains of S pneumonia and Legionella. This is a general statement that needs to be more specific on the empiric use of antimicrobials.

We agree with reviewer suggestion, it has been added in the revised manuscript

We agree with reviewer suggestion, it has been précised in the revised manuscript:i.e. a combination therapy with cephalosporin and macrolide or monotherapy with respiratory fluoroquinolone (83-85).

Methods

The study was conducted on ICU admitted patients, and they may not represent the study objectives rather than only severe cases that required ICU admission. So, study objectives need to be changed accordingly.

We agree with reviewer comment, it has been modified in the revised manuscript that it was all severe case of CAP hospitalized in ICU (line 91, 105).

It’s not clear too, how data were collected, was it a retrospective chart review?

It was a chart review of all patients hospitalized in ICU for severe CAP. It has been precized in the revised manuscript (line 104).

Statistical analysis

Line 5: First time mentioned abbreviation, the full name should be provided.

Also, the statement is not clear: “Risk factors of influenza PAC in bivariate analysis with P<0.1 were entered into a multivariate logistic regression analysis using backward selection with P <0.05”.

Has suggested by the Editor, the section on risk factors of Influenza CAP with multivariate analysis has been removed from the revised manuscript.

The reported incidence in the results section was different from the abstract and discussion, should be revised.

We apologize, it has been corrected in the revised manuscript

Table (1), abscess was mentioned, what does it refer to what type of abscess and which site?

It was pulmonary abscess, it has been added in the revised manuscript

Table (1), duration between symptoms and ICU admission, was it days?

It was days, it has been précised in the revised manuscript

Table (1), what does it mean of reporting the p-value as zero? (for influenza-like illness)

We apologize, this was an error, and these are P < 0.001. This has been corrected in the revised manuscript.

Table (1), how hazardous alcohol use was assessed, what definition was used, and how data were collected?

We have used The Alcohol Use Disorders Identiﬁcation Test Consumption (AUDIT-C) with a with a threshold value of 4 for men and 3 for women (Bradley KA, DeBenedetti AF, Volk RJ, Williams EC, Frank D, Kivlahan DR. AUDIT-C as a brief screen for alcohol misuse in primary care. Alcohol Clin Exp Res 2007 Jul;31(7):1208-17. doi: 10.1111/j.1530-0277.2007.00403.x). It has been added in the revised manuscript.

Table (1), what does footnote 2 refer to?

It was a spelling error and it has been removed from the revised manuscript 

Did the investigators collect any vaccination histories on influenza or pneumococcal vaccines among the study participants?

We apologize we dot not have this data

Table (3), footnote … “…. during flu cases” need revision.

What is “flu syndrome” mentioned in line 201?

We apologize during flu cases has been replaced by “Influenza CAP”. “Flu syndrome” was “Influenza-like illness” but it has been removed from the revised manuscript has it was in the multivariate analysis section.

Higher leukocyte counts mentioned in line 204 is not a risk for severe influenza as per the odds ratio presented.

We agree with reviewer comment. Moreover, it has been removed from the revised manuscript has it was in the multivariate analysis.

Line 212 need revision

We apologize, the line has been revised

Discussion

The second highest cause of mortality and years of life lost. Was there a difference between mortality and years of life lost?

We have deleted the second part of the sentence

in-ICU mortality in this study was lower than previously reported. Any specific reasons or contributing factors to be discussed?

It has been reported that mortality for CAP could reach 50% in patients hospitalized in ICU. Nevertheless, mortality in recent studies that evaluated severe CAP in-ICU patients was similar to that in our study: 22% in the study by Ferrer et al (Ferrer M, Travierso C, Cilloniz C, Gabarrus A, Ranzani OT, Polverino E, Liapikou A, Blasi F, Torres A. Severe community-acquired pneumonia: Characteristics and prognostic factors in ventilated and non-ventilated patients PLoS One 2018 Jan 25;13(1):e0191721. doi: 10.1371/journal.pone.0191721. eCollection 2018) and 27% in the study by Cavallazzi et al (Cavallazzi R, Furmanek S, Arnold FW, Beavin LA, Wunderink RG, Niederman MS, Ramirez JA. The Burden of Community-Acquired Pneumonia Requiring Admission to ICU in the United States. Chest 2020 Sep;158(3):1008-1016. doi: 10.1016/j.chest.2020.03.051. It has been added in the discussion of the revised manuscript (line 271-275)

Editing, line 253, “….. we find” should “found”

We apologize it has been corrected

What about the impact of empirical use of antimicrobials on the detection of organisms that have been reported in other reports.

The use of empirical antibiotics could theoretically have an impact on deep respiratory samples. But to our knowledge there is no study on the impact of one dose of antibiotic on the positivity of respiratory samples concerning CAP.

The investigators included a paragraph in the discussion about climate change and influenza types, although they didn’t provide a specific discussion or explored the relevance to the study.

We agree with reviewer suggestion, has also suggested by the Editor this section has been has been removed from the revised manuscript. Several studies find an association between seasons and the occurrence of CAP but the physio-pathological explanation of this association remains to be established. This explanatory paragraph was removed because it was unclear.

Conclusions should be revised to reflect the study findings.

We agree with reviewer suggestion, the conclusions have been modified.

Also, the recommendation of probabilistic antimicrobials and oseltamivir to every patient should be revised taking into consideration the potential risk for antimicrobial resistance and definition of severe pneumonia that may not be specific to be applicable in several care settings.

We agree with reviewer comment, we have removed from the revised manuscript the text concerning the treatment by oseltamivir.

Figure (2), worst survival was observed for cases with S pneumonia, this needs further discussion.

In the present study, the mortality rate in the subgroup of patients with S. pneumonia CAP was similar to that reported by Dupuis et al. They found that in-hospital mortality among the 1665 patients who required ICU admission for pneumococcal CAP was 22.8% on day 28 (Dupuis C , Sabra A, Patrier J, Chaize G, Saighi A, Féger C, Vainchtock A, Gaillat J, Timsit JF. Burden of pneumococcal pneumonia requiring ICU admission in France: 1-year prognosis, resources use, and costs. Crit Care 2021 Jan 10;25(1):24. doi: 10.1186/s13054-020-03442-z.). It has been added in the discussion section of the revised manuscript line 271-275). 

Figure (1) (which is also written as figure (2)), about rainfall and influenza, showed three peaks of rainfall but not clear of an association with the occurrence of influenza or pneumococcal infection, more elaboration in this regard is required.

There is an association between the dry season and the occurrence of influenza CAP. This association was not found with S. pneumoniae. Several studies find an association between seasons and the occurrence of Influenza CAP but the physio-pathological explanation of this association remains to be established. This explanatory paragraph was removed because it was unclear.

Reviewer #2: Severe community-acquired pneumonia in Reunion Island: Epidemiological, clinical and microbiological characteristics, 2016-2018 (Manuscript No. PONE-D-21-05049)

In their manuscript, de Mangou et al. present valuable data derived from a retrospective cohort of patients admitted and diagnosed with severe CAP at Reunion Island between 2016 and 2018. However, significant scarcities of methodology detailed below could be identified in the manuscript. Also, the main study results might be of more interest to the national readers of Reunion Island. In all, the Methods section needs rewriting by addition of more explicit details from the study protocol before the review process could be continued any further.

Remarks on methodology:

1. A major issue is that CAP was not defined properly (only criteria of severe CAP were stated), and details were not given on how cases of CAP were ascertained. A brief summary of in-hospital algorithms should be added to the manuscript. Also, did all the patients receive the same diagnostic (eg. imaging) and microbiological studies before the final diagnosis of CAP was arrived at?

We have better redefined our diagnosis criteria for CAP in the material and methods section (line XX-XX). Data were collected using a chart review of all patients hospitalized in ICU for severe CAP. It has been also precized in the revised manuscript (108-112).

2. What is a cytobacteriological examination? This needs further explaining.

We apologize, the sentence has been reworded. Bacteriological examination of respiratory samples were always performed (sputum samples on non-intubated patients and tracheal or bronchoalveolar sputum on intubated patients (122).

3. Did the patients have blood cultures taken?

Yes, all patients had blood cultures. It has been précised in the revised manuscript (line 122).

4. Are all patients with severe CAP hospitalized at the ICU ward at Reunion Island?

It has been précised that it was all patients with severe CAP and hospitalized in ICU that have been evaluated. Indeed, some patients are not admitted to the intensive care unit because of ethical considerations

5. How were eligible patients screened and included in the final cohort?

We performed a retrospective chart review of all patients diagnosed with severe CAP. It has been précised in the revised manuscript (line 104).

6. Was any patient follow-up seeked during the study period? If yes, how?

Follow up was seeked during in-hospital hospitalization on chart patients

7. How was data collection executed? What was the data source? How were missing data handled?

Data collection was retrospectively collected on patient charts. And a column with the item missing data has been added in the tables.

8. The primary outcome should be defined by a hard clinical end-point, eg. in-hospital all-cause mortality at day-14 or alike. "Clinical and microbiological characteristics" are not endpoints – they are characteristics.

We agree with reviewer suggestion the primary outcome has been modified in the revised manuscript (line 137)

9. There is a contradiction in the description of statistical methods: an univariate pre-screening was done (line 133), but after that, only the "most clinically relevant factors" were entered into multivariate analysis (line 135)? This needs to be elaborated.

We agree with reviewer comment, has suggested by the editor but the multivariate analysis has been removed from the revised manuscript

10. Adherence to the STROBE Statement is highly encouraged during rewriting.

We agree with reviewer comment, adherence to the STROBE checklist has been checked in the revised manuscript (missing data in tables…) and a statement has been added (101).

11. I suggest using the term bronchoalveolar lavage instead of bronchoalveolar sputum.

We apologize, it has been corrected in the revised manuscript 

12. I suggest that the term "adult patients" should be emphasized.

We agree with reviewer comment, it has been specified in the revised manuscript (104)

13. The name of the American Thoracic Society should be corrected.

We apologize, it has been corrected in the

---

## [Decision Letter · Decision Letter 1]

4 Jan 2022

PONE-D-21-05049R1Severe community-acquired pneumonia in Reunion Island:

Epidemiological, clinical and microbiological characteristics, 2016-2018PLOS ONE

Dear Dr. Allou,

Thank you for submitting your manuscript to PLOS ONE. After careful consideration, we feel that it has merit but does not fully meet PLOS ONE’s publication criteria as it currently stands. Therefore, we invite you to submit a revised version of the manuscript that addresses the points raised during the review process.

The revised manuscript has addressed most of the reviewers’ comments, however, there are several issues that still need further clarification, please see reviewer’s comments and specific comments below.  In addition, the quality of the language still needs to be improved, We suggest you thoroughly copyedit your manuscript for language usage, spelling, and grammar. If you do not know anyone who can help you do this, you may wish to consider employing a professional scientific editing service.

Specific comments:

Line 78 – 80:  This statement needs reference(s).Line 83 & 163 :  Change “*S. pneumonia*” to “*S. pneumoniae*”, “*H. influenza*” to “*H. influenzae*”.Line 100:  this statement needs further clarification.Line 122 – 124:  Needed to specify the methodology used? Culture or other methods?Line 252:  Change “Acinetobacter” to “*Acinetobacter*”.Discussion:  Use either oseltamivir or Oseltamivir:  Be consistent. Please submit your revised manuscript by Feb 18 2022 11:59PM. If you will need more time than this to complete your revisions, please reply to this message or contact the journal office at plosone@plos.org. Please include the following items when submitting your revised manuscript:A rebuttal letter that responds to each point raised by the academic editor and reviewer(s). You should upload this letter as a separate file labeled 'Response to Reviewers'.A marked-up copy of your manuscript that highlights changes made to the original version. You should upload this as a separate file labeled 'Revised Manuscript with Track Changes'.An unmarked version of your revised paper without tracked changes. You should upload this as a separate file labeled 'Manuscript'.If applicable, we recommend that you deposit your laboratory protocols in protocols.io to enhance the reproducibility of your results. Protocols.io assigns your protocol its own identifier (DOI) so that it can be cited independently in the future. For instructions see: https://journals.plos.org/plosone/s/submission-guidelines#loc-laboratory-protocols. Additionally, PLOS ONE offers an option for publishing peer-reviewed Lab Protocol articles, which describe protocols hosted on protocols.io. Read more information on sharing protocols at https://plos.org/protocols?utm_medium=editorial-email&utm_source=authorletters&utm_campaign=protocols.

We look forward to receiving your revised manuscript.

Kind regards,

Baochuan Lin, Ph.D.

Academic Editor

PLOS ONE

Journal Requirements:

Reviewers' comments:

Reviewer's Responses to Questions

**Comments to the Author**

1. If the authors have adequately addressed your comments raised in a previous round of review and you feel that this manuscript is now acceptable for publication, you may indicate that here to bypass the “Comments to the Author” section, enter your conflict of interest statement in the “Confidential to Editor” section, and submit your "Accept" recommendation.

Reviewer #1: All comments have been addressed

Reviewer #2: All comments have been addressed

2. Is the manuscript technically sound, and do the data support the conclusions?

Reviewer #1: Yes

Reviewer #2: Yes

3. Has the statistical analysis been performed appropriately and rigorously? 

Reviewer #1: Yes

Reviewer #2: Yes

4. Have the authors made all data underlying the findings in their manuscript fully available?

Reviewer #1: Yes

Reviewer #2: Yes

5. Is the manuscript presented in an intelligible fashion and written in standard English?

Reviewer #1: Yes

Reviewer #2: Yes

6. Review Comments to the Author

Reviewer #1: Thank you for revising your manuscript. However, there are some points that I think need a revision.

1. Conclusion in the abstract and at end of the manuscript that melioidosis and Acinetobacter app should be considered in patients returning from Madagascar, this conclusion is based on only a few number of cases and shouldn't be generalized especially an adjusted regression analysis was not performed.

2. In the Materials and methods, the investigators mentioned that written information notice about the process of data collection was obtained, but they didn't report about severely ill patients and those on ventilators or not fully conscious, how that process was complete.

3. In the Material and methods, third paragraph: a reference on the CAP clinical definition is required.

4. In the results section: The acute physiology score was first mentioned with no explanation of its definition, components, or reference. The same was in Table1 for CURB-65.

5. Table 2: Biological characteristics should be changed into laboratory findings.

6. Discussion should highlight that severity of influenza-CAP may be associated with patients' underlying conditions not directly related to the influenza infection itself compared to other infections. This can be evidenced if a multivariate regression was performed.

7. Kruskal-Walis test was mentioned in the statistical analysis, although not shown or used in the results.

Reviewer #2: In their revised manuscript, de Mangou et al. addressed all the methodological remarks. A reconsideration of publication could be recommended.

7. PLOS authors have the option to publish the peer review history of their article (what does this mean?). If published, this will include your full peer review and any attached files.

Reviewer #1: **Yes: **Fayssal Farahat

Reviewer #2: No

---

## [Author Response · Author response to Decision Letter 1]

18 Jan 2022

Nicolas Allou, MD January, 14th, 2022

CHU Félix Guyon

Réanimation Polyvalente

Allée des Topazes, 97400, Saint Denis, France

Phone: +2 62 90 56 94

Fax: +2 62 90 66 93

E-mail: nicolas.allou@hotmail.fr
nicolas.allou@chu-reunion.fr

Dear Pr. Baochuan Lin,

Thank you for your letter concerning our manuscript entitled “Severe community-acquired pneumonia at Reunion Island: Epidemiological, clinical and microbiological characteristics, 2016-2018" (PONE-D-21-05049R1).

The comments were stimulating and challenging. We have therefore deeply modified the manuscript, taking all of the suggestions into account. Changes in the manuscript are in red font.

The manuscript has not and will not be offered elsewhere for possible publication, as long as it is under PLOS ONE consideration.

All relevant data are within the manuscript and its Supporting Information files

Ethics statement and consents from participants have been specified in the revised manuscript

The authors received no specific funding for this work.

We give a point-by-point response to the reviewers (see below) 

We hope that these revisions have improved the manuscript.

Nicolas ALLOU, MD 

COMMENTS TO AUTHOR:

Reviewer #1: Thank you for revising your manuscript. However, there are some points that I think need a revision.

1. Conclusion in the abstract and at end of the manuscript that melioidosis and Acinetobacter app should be considered in patients returning from Madagascar, this conclusion is based on only a few number of cases and shouldn't be generalized especially an adjusted regression analysis was not performed.

We agree with reviewer comment, it has been deleted from the conclusion of the revised manuscript (line 68).

2. In the Materials and methods, the investigators mentioned that written information notice about the process of data collection was obtained, but they didn't report about severely ill patients and those on ventilators or not fully conscious, how that process was complete.

We have clarified that in the revised manuscript. Information notice about the process of data collection was obtained from their legally authorized representative in case of patients unable to obtain the information (line 99).

3. In the Material and methods, third paragraph: a reference on the CAP clinical definition is required.

We agree with reviewer comment, it has been added in the revised manuscript line 110).

4. In the results section: The acute physiology score was first mentioned with no explanation of its definition, components, or reference. The same was in Table1 for CURB-65.

We agree with reviewer comment, references have been added in the revised manuscript (Lim WS, van der Eerden MM, Laing R, et al. (2003). "Defining community acquired pneumonia severity on presentation to hospital: an international derivation and validation study"/ A New Simplified Acute Physiology Score (SAPS II) Based on a European/North American Multicenter Study. JAMA. 1993).

5. Table 2: Biological characteristics should be changed into laboratory findings.

We apologize, it has been corrected in the revised manuscript 

6. Discussion should highlight that severity of influenza-CAP may be associated with patients' underlying conditions not directly related to the influenza infection itself compared to other infections. This can be evidenced if a multivariate regression was performed.

We agree with reviewer comment. We found that patients in the group with influenza CAP had higher mortality rate than patients with non-influenza CAP following univariate analysis. Nevertheless, patients with influenza CAP had more frequent comorbidities (age over 65 years old, asthma, diabetes mellitus…) and there is no multivariate analysis performed. It has been added in the revised manuscript (line 275).

7. Kruskal-Walis test was mentioned in the statistical analysis, although not shown or used in the results.

We agree with reviewer comment. It has been deleted from the revised manuscript.

Reviewer #2: In their revised manuscript, de Mangou et al. addressed all the methodological remarks. A reconsideration of publication could be recommended.

Specific comments:

1. Line 78 – 80: This statement needs reference(s).

We agree with reviewer comment. References have been added in the revised manuscript

Microbial Etiology of Pneumonia: Epidemiology, Diagnosis and Resistance Patterns. Cilloniz et al. 2016

But clearly, it can’t be generalized because there are differences between causative organisms encountered in the tropics compared with temperate climates. Due to both environmental and socio- economic factors, diseases such as melioidosis, leptospirosis and tuberculosis are more widespread in the tropics. For example, observation of the aetiologies of community-acquired pneumonia (CAP) in an Asian outpatient setting showed that the most common isolates were Chlamydophila pneumoniae, followed by Mycoplasma pneumoniae and Streptococcus pneumoniae. Lim_et_al-2018-Respirology_pneumonia in the tropics.

2. Line 83 & 163 : Change “S. pneumonia” to “S. pneumoniae”, “H. influenza” to “H. influenzae”.

We apologize, it has been corrected in the revised manuscript 

3. Line 100: this statement needs further clarification.

We agree with reviewer comment, it has been clarified in the revised manuscript (line 98).

4. Line 122 – 124: Needed to specify the methodology used? Culture or other methods?

We agree with reviewer comment it has been specified in the revised manuscript that identification was performed with Gram staining followed by culturing with definite microorganism identification. Alternatively, identification was performed with matrix-assisted laser desorption ionization time-of-flight mass spectrometry.

5. Line 252: Change “Acinetobacter” to “Acinetobacter”.

We apologize, it has been corrected in the revised manuscript 

6. Discussion: Use either oseltamivir or Oseltamivir: Be consistent.

We apologize, it has been corrected in the revised manuscript

---

## [Decision Letter · Decision Letter 2]

9 Feb 2022

PONE-D-21-05049R2Severe community-acquired pneumonia in Reunion Island:

Epidemiological, clinical and microbiological characteristics, 2016-2018PLOS ONE

Dear Dr. Allou,

Thank you for submitting your manuscript to PLOS ONE. After careful consideration, we feel that it has merit but does not fully meet PLOS ONE’s publication criteria as it currently stands. Therefore, we invite you to submit a revised version of the manuscript that addresses the points raised during the review process.

Reviewer and I agreed that the revised manuscript is scientifically sound, however, the language still needs improvement, there are still quite a few of typos and grammatic error throughout the manuscript (see reviewer's comments and specific comments below as example.  We suggest you thoroughly copyedit your manuscript for language usage, spelling and grammar.  If you do not know who can help you do this, you may wish to consider employing a professional scientific editing service.

Whilst you may use any professional scientific editing service of your choice, PLOS has partnered with both American Journal Experts (AJE) and Editage to provide discounted services to PLOS authors. Both organizations have experience helping authors meet PLOS guidelines and can provide language editing, translation, manuscript formatting, and figure formatting to ensure your manuscript meets our submission guidelines. To take advantage of our partnership with AJE, visit the AJE website (http://learn.aje.com/plos/) for a 15% discount off AJE services. To take advantage of our partnership with Editage, visit the Editage website (www.editage.com) and enter referral code PLOSEDIT for a 15% discount off Editage services. If the PLOS editorial team finds any language issues in text that either AJE or Editage has edited, the service provider will re-edit the text for free. Specific comments:1. Line 79, change etc to etc.

2. Line 83, change *i.e.*a to *i.e*. a

3. Line 96, change N° to No.

4. Line 98, delete "about the process of data collection"

5. Line 265, change S. pneumonia to S. pneumoniae

6. Line 276, change "...diabetes mellitus…" to "...diabetes mellitus, etc."

 Please submit your revised manuscript by Mar 26 2022 11:59PM. If you will need more time than this to complete your revisions, please reply to this message or contact the journal office at plosone@plos.org. Please include the following items when submitting your revised manuscript:A rebuttal letter that responds to each point raised by the academic editor and reviewer(s). You should upload this letter as a separate file labeled 'Response to Reviewers'.A marked-up copy of your manuscript that highlights changes made to the original version. You should upload this as a separate file labeled 'Revised Manuscript with Track Changes'.An unmarked version of your revised paper without tracked changes. You should upload this as a separate file labeled 'Manuscript'.If applicable, we recommend that you deposit your laboratory protocols in protocols.io to enhance the reproducibility of your results. Protocols.io assigns your protocol its own identifier (DOI) so that it can be cited independently in the future. For instructions see: https://journals.plos.org/plosone/s/submission-guidelines#loc-laboratory-protocols. Additionally, PLOS ONE offers an option for publishing peer-reviewed Lab Protocol articles, which describe protocols hosted on protocols.io. Read more information on sharing protocols at https://plos.org/protocols?utm_medium=editorial-email&utm_source=authorletters&utm_campaign=protocols.

We look forward to receiving your revised manuscript.

Kind regards,

Baochuan Lin, Ph.D.

Academic Editor

PLOS ONE

Journal Requirements:

Reviewers' comments:

Reviewer's Responses to Questions

Comments to the Author

1. If the authors have adequately addressed your comments raised in a previous round of review and you feel that this manuscript is now acceptable for publication, you may indicate that here to bypass the “Comments to the Author” section, enter your conflict of interest statement in the “Confidential to Editor” section, and submit your "Accept" recommendation.

Reviewer #1: (No Response)

2. Is the manuscript technically sound, and do the data support the conclusions?

Reviewer #1: (No Response)

3. Has the statistical analysis been performed appropriately and rigorously? 

Reviewer #1: (No Response)

4. Have the authors made all data underlying the findings in their manuscript fully available?

Reviewer #1: Yes

5. Is the manuscript presented in an intelligible fashion and written in standard English?

Reviewer #1: Yes

6. Review Comments to the Author

Reviewer #1: (No Response)

7. PLOS authors have the option to publish the peer review history of their article (what does this mean?). If published, this will include your full peer review and any attached files.

Do you want your identity to be public for this peer review? For information about this choice, including consent withdrawal, please see our Privacy Policy.

Reviewer #1: No

---

## [Author Response · Author response to Decision Letter 2]

23 Mar 2022

Nicolas Allou, MD March, 22th, 2022

CHU Félix Guyon

Réanimation Polyvalente

Allée des Topazes, 97400, Saint Denis, France

Phone: +2 62 90 56 94

Fax: +2 62 90 66 93

E-mail: nicolas.allou@hotmail.fr
nicolas.allou@chu-reunion.fr

Dear Pr. Baochuan Lin,

Thank you for your letter concerning our manuscript entitled “Severe community-acquired pneumonia at Reunion Island: Epidemiological, clinical and microbiological characteristics, 2016-2018" (PONE-D-21-05049R2).

As recommended, the manuscript has been revised by an English translator specialized in medical English (Ariane Dorval) and has been greatly improved. Changes in the manuscript are in red font.

The manuscript has not and will not be offered elsewhere for possible publication, as long as it is under PLOS ONE consideration.

All relevant data are within the manuscript and its Supporting Information files

Ethics statement and consents from participants have been specified in the revised manuscript

The authors received no specific funding for this work.

We give a point-by-point response to the reviewers (see below) 

We hope that these revisions have improved the manuscript.

Nicolas ALLOU, MD

---

## [Editor Report · Decision Letter 3]

5 Apr 2022

Severe community-acquired pneumonia in Reunion Island:

Epidemiological, clinical and microbiological characteristics, 2016-2018

PONE-D-21-05049R3

Dear Dr. Allou,

We’re pleased to inform you that your manuscript has been judged scientifically suitable for publication once you correct the following:  Line 55: delete "consecutive", line 78" suggest changing from "they recommend" to "the recommendations are", and line 273, italicize "S. aureus" to "*S. aureus*', and will be formally accepted for publication once it meets all outstanding technical requirements. 

Kind regards,

Baochuan Lin, Ph.D.

Academic Editor

PLOS ONE
---

## [Editor Report · Acceptance letter]

7 Apr 2022

PONE-D-21-05049R3 

Severe community-acquired pneumonia in Reunion Island:
Epidemiological, clinical, and microbiological characteristics, 2016-2018 

Dear Dr. Allou:

I'm pleased to inform you that your manuscript has been deemed suitable for publication in PLOS ONE. Congratulations! Your manuscript is now with our production department. 

Kind regards, 

on behalf of

Dr. Baochuan Lin 

Academic Editor

PLOS ONE